# Assessing the Impact of Irrigation and Biostimulants on the Yield and Quality Characteristics of Two Different St. John’s Wort Cultivars in Their Second Growing Season

**DOI:** 10.3390/plants13243573

**Published:** 2024-12-21

**Authors:** Athina Tegou, Kyriakos D. Giannoulis, Elias Zournatzis, Savvas Papadopoulos, Dimitrios Bartzialis, Nikolaos G. Danalatos, Eleni Wogiatzi-Kamvoukou

**Affiliations:** 1Department of Agrotechnology, University of Thessaly, 41500 Larissa, Greece; ategou@uth.gr (A.T.); iliasz@hotmail.gr (E.Z.); psavvas1@yahoo.gr (S.P.); wogiatzi@uth.gr (E.W.-K.); 2Department of Agriculture, Crop Production and Rural Environment, University of Thessaly, 38446 Volos, Greece; dbartz@uth.gr (D.B.); danal@uth.gr (N.G.D.)

**Keywords:** *Hypericum perforatum*, yield, essential oil, aromatic–medicinal plants, biostimulants, irrigation

## Abstract

The perennial species *Hypericum perforatum*, commonly known as St. John’s Wort, is well regarded for its medicinal attributes, particularly its strong anti-inflammatory and antidepressant effects. *Hypericum perforatum* L., commonly known as balsam, is extensively employed in both traditional and contemporary medicine due to its biological properties, although the plant’s medicine distribution is limited to Europe and Asia. This study pioneers the investigation of *Hypericum perforatum* cultivation in a Mediterranean country, specifically Greece, focusing on the effects of irrigation and biostimulants of two distinct genotypes on quantitative (height, drug yield, essential oil yield) and qualitative (essential oil content and composition) characteristics. A field trial was conducted at the experimental farm of the Agrotechnology Department at the University of Thessaly, located in the Larissa region. This study investigated various testing varieties under different irrigation levels and biostimulant applications. The results underscore the importance of customized irrigation and biostimulant strategies in improving yield and quality during the second growing season, establishing a foundation for sustainable agricultural progress. Notably, irrigated treatments significantly increased plant height, dry biomass yield, and essential oil production per hectare. Specifically, the essential oil yields for irrigated treatments were nearly double those of rainfed treatments, with 219 kg/ha for rainfed and 407 kg/ha for irrigated. The genotype played a crucial role in influencing production potential, height, flowering, and essential oil composition, with one variety demonstrating biennial blooming and modified essential oil compounds. While irrigation positively impacted yield, it also reduced certain essential oil compounds while increasing β-pinene content. The effects of biostimulants varied based on their composition, with some enhancing and others diminishing essential oil content. Notably, the biostimulant containing algae with auxin and cytokinin (B2) proved to be the most effective in improving the therapeutic profile. This study offers valuable insights into the cultivation of *H. perforatum* in a Mediterranean climate, highlighting the necessity for ongoing research into native populations, irrigation levels, biostimulants, fertilization, and other factors that affect crop yield and quality characteristics.

## 1. Introduction

*Hypericum perforatum* L., widely recognized as St. John’s Wort, stands out as the most prominent perennial species within the Hypericaceae family, found across Central and Eastern Europe, Asia, North Africa, and North America [1,2]. This plant has a long history of medicinal use, spanning over two millennia [3]. Driven by the demand for natural therapeutic agents and the beneficial biological properties of *Hypericum perforatum* L., it is frequently incorporated into medical practices [4,5,6,7,8]. Its extracts are noted for various applications, including antiviral, antimicrobial, anti-inflammatory, and antitumor properties [9], as well as their effectiveness in alleviating symptoms of depression [10]. According to pharmacopoeial standards, the raw material derived from *Hypericum perforatum* consists of the flowering tops of the shoots (*Hyperici herba*), which must meet specific compound content requirements [11].

Numerous studies have compared the chemical compositions of native and commercial samples [12,13]. For instance, ref. [14] examined how the origin and biological source influence the chemical makeup and biological properties of various St. John’s Wort species. The extracts from all samples demonstrated significant antioxidant potential; however, in most instances, the indigenous samples exhibited stronger effects than the original source. Variations in the levels of phenolic compounds, particularly flavonoids, were linked to the source of the drug, which impacted the observed activities. Nevertheless, further research is necessary to investigate how commercial varieties adapt to different climatic conditions and the subsequent effects on their phytochemical profiles.

In Europe, the main production of this crop takes place in Belarus, Germany, Italy, Poland, Romania, Switzerland, and Siberia. A significant portion of the total supply of *Hypericum* comes from the collection of wild plants, which has resulted in a troubling decrease in their natural populations [15,16]. Although the cultivation of *Hypericum perforatum* offers a lucrative opportunity for farmers, the lack of standardized agricultural practices for its growth poses a challenge. Furthermore, slow growth rates, low yields, and variable quality present considerable obstacles to its broader cultivation [17].

St. John’s Wort is a perennial shrub that can grow between 30 and 100 cm tall and is capable of reproducing through both vegetative and sexual means. Notable physical characteristics of *Hypericum perforatum* include its taproot system, woody stems, and rhizomes. The leaves are distinguished by translucent glands scattered throughout the lamina, and the plant produces cymes of yellow flowers that mature into dehiscent capsules filled with seeds [18].

The quality of St. John’s Wort can differ due to various factors, including the specific sub-species or varieties, the environmental conditions where it is grown, the timing of the harvest [19], and the cultivation techniques employed. Research indicates that there are notable genotypic and phenotypic variations among different St. John’s Wort populations [20]. To identify these genetic distinctions, scientists conduct morphological and biochemical analyses [21]. However, there is limited information on the cultivation and production of *Hypericum*, despite extensive documentation of the medicinal benefits associated with its aerial parts [22,23,24].

As mentioned above, St. John’s Wort is primarily recognized for its applications as a food additive and in medicinal contexts. Consequently, it is crucial to explore the potential for large-scale production utilizing organic farming methods. In this context, the incorporation of biostimulants—natural compounds or microbial products—merits consideration. These substances can greatly improve plant growth and their inherent processes when applied [25].

In recent years, seaweed extracts have gained increased attention as a category of biostimulants, largely due to various research studies [26,27]. As noted by [28], these seaweed-derived biostimulants contribute to improved crop yields, enhanced root development, flowering, resilience to stress, and better nitrogen uptake.

As previously noted, similar to many medicinal plants, there are no established cultivation protocols akin to those for traditional crops, despite their extensive use over the years. The impact of cultivation methods, including those applicable to organic farming, remains largely unknown for any species within the aromatic and medicinal plant category. Consequently, this research focused on cultivating two recognized varieties of *Hypericum perforatum* L. in Greece, employing both dry and fully irrigated treatments to explore the potential for utilizing all types of land, including those lacking water resources. Additionally, the study sought to investigate the application of biostimulants and their influence on overall performance regarding yield and quality during the second growing season, particularly since one of the varieties blooms biennially. Notably, biostimulants can be employed within organic production systems, underscoring their significance when these crops are intended as raw materials for pharmaceutical applications.

## 2. Materials and Methods

A comprehensive experiment that involved different varieties, different irrigation levels, and different biostimulant treatments was conducted. The study was designed to assess their effects on growth metrics, yield outputs, and the qualitative attributes of the St. John’s Wort varieties, including essential oil composition.

### 2.1. Experimental Site, Design, and Management

In May 2022, a field experiment was conducted at the experimental farm of the Agrotechnology Department at the University of Thessaly, situated in the Larissa region of Greece (latitude 39°37′34.0″ N, longitude 22°22′52.4″ E). The objective of this study was to examine the impact of irrigation and biostimulant application on the overall performance of two well-known varieties (Topaz and Taubertal) of St. John’s Wort, focusing on aspects such as crop development, drug yield, essential oil content, and the quality of essential oil within a typical Mediterranean soil–climatic setting.

A factorial split-split-plot design was used with six replicates (blocks) and twelve plots per replication. The main factor was the different testing varieties (V1: Topaz and V2: Taubertal). The main sub-factor was the different irrigation levels (I1: rainfed and I2: 100% reference evapotranspiration (ETo); using a drip irrigation system while the irrigation schedule was determined according to the Class A evaporation pan method [29] and irrigation was conducted on a weekly basis, aiming to replicate the actual conditions experienced by farmers during their irrigation practices). The main sub-sub-factor was the different biostimulant applications (B1: control; B2: seaweeds; auxin 1.0 mg/lt and cytokinin 0.031 mg/lt from Ecklonia maxima; application rate of 0.8 L per hectare per year; and B3: a specific formulation including 10% *w*/*v* of amino acids, 11.3% *w*/*v* of pure protein, 22% *w*/*v* of sucrose, and 10% *w*/*v* of UV filter; application rate of 1.6 L per hectare per year). The biostimulants were applied annually prior to the onset of flowering (in the final ten days of May and right after the prevailing rainfall), utilizing the specified amounts and employing a micro electric sprayer for precise application.

Certified seedlings of *Hypericum perforatum* L. (varieties Topaz and Taubertal) were transplanted into the field at a spacing of 60 cm between plants and 60 cm between rows. Each plot measured 2 m × 1.70 m and contained 12 plants in 3 rows. The findings presented pertain to the second growing season, during which both varieties exhibited blooming, noting that one of the varieties blooms every other growing year.

It is important to note that both varieties of plants re-germinated within the first ten days of March; however, there was a variation in the onset of flowering between them. Topaz began to flower earlier than Taubertal, resulting in an earlier harvest for Topaz. Consequently, Topaz was harvested around June 10, while Taubertal was harvested 20 days later, on June 29.

### 2.2. Soil Physical–Chemical Characteristics

The soil of the experimental field is a sandy loam (SL) and is characterized as semi-fertile to unfertile (organic matter of 1.3% at a depth of 0–30 cm). It is an alkaline nature (pH 7.6), with a bulk specific gravity of 1.3 g cm^−3^ and electrical conductivity of 0.2 ms/cm.

### 2.3. Plant Growing Characteristics and Yield

The height, flowering height, and overall yield were assessed at the ideal harvesting stage, which corresponds to full flowering, a stage known to maximize essential oil content. To prevent any border effects, the harvested flowers from each plot were taken from the inner row, specifically two plants per plot.

The aerial parts (flowers and upper leaves) were collected, weighed, and promptly transported to the laboratory, where they were air-dried in a shaded, well-ventilated area at ambient temperature (25–28 °C) until a constant weight was achieved. The dry weight was recorded for each plot to calculate the dry biomass yield per hectare and the distillation process followed.

Manual weeding was conducted every two weeks to manage weed competition, while there were no notable occurrences of pests or diseases; therefore, no pesticides were used (Figure 1).

### 2.4. Essential Oil Yield and Quality Characteristics

The dried samples were processed in a laboratory mill to achieve a consistent particle size of around 1 mm. Thereafter, the essential oil content was assessed using a Clevenger-type distillation apparatus. A total of 10 g of the dried plant material was subjected to hydro distillation with 250 mL of distilled water for 2.5 h. This process was conducted three times for each sample, and the essential oil content was calculated based on the dry weight of the plant material. The extracted essential oils were measured volumetrically and stored in amber glass vials in a refrigerator at 4 °C until further analysis [30,31]. Following storage, the essential oils were analyzed using gas chromatography coupled with mass spectrometry (GC–MS) on a fused silica DB-5 column. The relative content of each compound was determined as a percentage of the total chromatographic area, and the results are presented as the mean percentage of three replicates [31,32]. Compound identification was achieved by comparing retention indices (RIs) relative to n-alkanes (C7–C22) with literature data and matching spectra against mass spectrometry libraries (NIST 98, Willey) [33].

Moreover, the yield of the essential oil was calculated by multiplying the essential oil concentration by the dry biomass yield per hectare.

### 2.5. Meteorological Data and Statistical Analysis

Weather data, encompassing air temperature, radiation, humidity, wind speed, precipitation, and the class-A pan evaporation rate, were gathered from an automatic meteorological station situated near the experimental field. The data collected were subjected to an analysis of variance (ANOVA) for all measured and derived variables at designated time intervals, employing the GenStat statistical software (7th Edition). The differences in means for both the main effects and interaction effects were assessed using the LSD_0.05_ (least significant difference at 5%) test criteria [34]. This statistical evaluation enabled a thorough examination of the data, confirming that any observed differences among the studied variables were statistically significant and not due to random variation.

## 3. Results and Discussion

### 3.1. Meteorological Data

The Mediterranean climate characteristic of the research area is recognized for its warm summers and mild winters. In the experimental year under investigation, the average air temperature during the crop re-emergence phase ranged from 2.1 to 21.8 °C (average air temperature 11.9 °C). Throughout the crop growth period (re-emergence–beginning of flowering), the maximum air temperature reached 29.3 °C, while the minimum was recorded at 14 °C (average air temperature 20.8 °C; Figure 2). Thereafter, during the growing stages between the beginning of flowering until harvest, the maximum air temperature reached 29.3 °C, while the minimum was recorded at 14 °C (average air temperature 20.8 °C; Figure 2).

Precipitation is another vital environmental factor that greatly influences crop development and yield. In 2023, the region experienced 194.2 mm of rainfall during the growing season (re-emergence–harvest). Specifically, between this period, the monthly rainfall totals were recorded as 60.6 mm, 39 mm, 62.4 mm, and 0 mm for March to June, respectively.

These data highlight the occurrence of drought conditions during the summer months, particularly from April to June (Figure 2).

### 3.2. Plant Characteristics and Yield

As indicated in Table 1, there were statistically significant differences among the varieties examined for all three characteristics: plant diameter, inflorescence height, and plant height. Furthermore, the irrigation factor also demonstrated statistically significant differences concerning inflorescence height and plant height.

Specifically, Variety 1 (Topaz) demonstrated a statistically significant superiority in both flowering height and overall plant height, with its final measurement exceeding that of Variety 2 by 15 cm. Conversely, Variety 2 (Taubertal) resulted in plants exhibiting a larger crown diameter, with a difference of nearly 5 cm, suggesting a greater capacity for spatial expansion (Table 1).

Concerning the irrigation factor, it was anticipated that the fully irrigated treatment (100% ETo) would produce the greatest values for both inflorescence height and overall plant height in comparison to the rainfed treatment. This anticipation is corroborated by meteorological data, which reveal a combination of recorded rainfall and elevated temperatures throughout the crop growth period. Notably, the total rainfall measured was 194.2 mm, while evaporation during the same timeframe reached 446.3 mm (data not presented). As illustrated in Table 1, the irrigation treatment led to plants that were, on average, 15 cm taller, with an average height of approximately 80 cm.

Finally, a statistically significant difference in plant height was observed regarding the interaction between the varieties and irrigation methods. Notably, the fully irrigated Variety 1 (Topaz) demonstrated the greatest plant height, measuring 90.6 cm, compared to the other treatments analyzed (Table 1).

The statistical analysis presented in Table 2 indicates that significant differences in production metrics, specifically fresh and dry weight, were observed solely among the examined cultivars.

In particular, Variety 1 (Topaz) demonstrated the highest production levels, achieving fresh and dry weights of 8371 kg and 3137 kg per hectare, respectively. These figures are nearly double those of the second variety, Taubertal, as shown in Table 2.

Irrigation and biostimulants did not demonstrate a statistically significant impact on crop yield, as indicated in Table 2. The average yields for the irrigation factor and biostimulants, irrespective of the variety, were approximately 6500 kg per hectare for fresh weight and 2400 kg per hectare for dry weight.

Finally, the interaction analysis of the three factors did not demonstrate a statistically significant impact on crop yield, while it indicated that Variety 1 (Topaz) attained maximum dry weight production under full irrigation (100% ETo) conditions without the use of any biostimulant (control), resulting in a yield of 3500 kg per hectare.

### 3.3. Essential Oil Yield and Quality Characteristics Measurements

According to the ANOVA table below (Table 3), the essential oil content does not appear to vary significantly among the different varieties of St. John’s Wort. However, it is influenced by the irrigation and fertilization treatments, as well as their interactions.

Irrigation appears to have a beneficial impact on the essential oil percentage, nearly doubling its content. In terms of biostimulants, B2 seems to enhance the oil content significantly, whereas B3 shows a detrimental effect when compared to the control (Table 3).

According to the essential oil content analysis, Variety 2 (Taubertal) has the highest essential oil concentration at 0.235% when it is subjected to the irrigation regime and treated with Biostimulant 2 (B2: seaweeds, auxin at 1.0 mg/lt, and cytokinin at 0.031 mg/lt derived from Ecklonia maxima).

To calculate the essential oil production per hectare in kilograms, the percentage essential oil content must be multiplied by the yield of the crop (dry weight per hectare, see Table 2). The result will represent the yield of essential oil per hectare.

Table 3 illustrates the statistical evaluation of essential oil production concerning the study factors analyzed, which include varieties, irrigation, and biostimulants. The results indicate that only two of the three factors exhibit a statistically significant impact on essential oil yield per hectare. Specifically, when comparing the varieties, Topaz (V1) shows a statistically significant superiority over Taubertal (V2), with essential oil production capacities of approximately 397 kg per hectare and 229 kg per hectare, respectively.

The irrigation factor appears to have a similar impact, as treatments with irrigation led to higher yields. In particular, irrigated treatments (I2) can yield approximately 407 kg of essential oils per hectare, in contrast to the 219 kg per hectare observed in the rainfed treatments (I1).

The interaction of irrigation factor with biostimulants demonstrated a statistically significant advantage for the irrigated treatments that utilized the combination of the two biostimulants, B2 and B3, compared to all other treatments. This approach resulted in essential oil production reaching as high as 468 kg per hectare (I2B2; Table 3). In the case of the interaction of varieties with biostimulants, a statistically significant benefit was observed for the Topaz variety (V1), irrespective of the type of biostimulant used, achieving an essential oil yield of 407 kg per hectare (V1B1, Table 3).

The expected result, considering the previously mentioned factors, indicates that the treatment resulting in the greatest production of essential oil, amounting to 566 kg per hectare, is the irrigated treatment (I2) of the Topaz variety (V1) when paired with Biostimulant 3 (B3). This finding demonstrates a numerical advantage rather than a statistically significant difference when compared to the other biostimulant treatments (i.e., V1I2B1 is roughly equivalent to V1I2B2 and V1I2B3, as shown in Table 3).

The analysis of the essential oil obtained from the experimental treatments, conducted using gas chromatography coupled with mass spectrometry (GC–MS), revealed not only the primary compounds hypericone and pseudo-hypericin (which are not presented in the current study) but also identified an additional 14 compounds present in concentrations exceeding 1%. These compounds include 2-Methyl-Decane, 2-Methyl-Octane, (E)-β-Farnesene, α-Pinene, Benzyl Benzoate, 2-β-Funebrene, γ-Muurolene, (E)-β-Ocimene, Oxide-Caryophyllene, β-Pinene, (E)-Caryophyllene, γ-Himachelene, δ-Cadinene, and Spathulenol. The effects of the studied factors are detailed in the Anova table provided below (Table 4).

It appears that all three factors—variety, irrigation levels, and biostimulants—have a statistically significant impact, depending on the compounds analyzed. Among the thirteen compounds studied, nine (2-methyl-decane, (ε)-β-phamesene, benzyl benzoate, 2-β-Funebrene, γ-Muurolene, osimene (ε-β), Caryophyllene oxide, (E)-Caryophyllene, and spathulenol) show a statistically significant effect related to the various biostimulants. Additionally, seven compounds (γ-Muurolene, Caryophyllene oxide, β-pinene, (E)-Caryophyllene, δ-Cadinene, and spathulenol) are significantly affected by the different irrigation levels, while seven compounds ((ε)-β-phamesene, α-pinene, benzyl benzoate, 2-β-Funebrene, Caryophyllene oxide, β-pinene, (E)-Caryophyllene, and Spathulenol) demonstrate significant variation across the different varieties.

All of the examined factors influence the composition of the essential oils and the percentage of each component. It is essential to note that the Taubertal variety does not contain “(E)-β-Farnesene” and “2-β-Funebrene”, while there is a significant reduction (over 50%) in the levels of “Oxide-Caryophylene”, “β-Pinene”, and “Benzyl Benzoate.” In contrast, the content of “α-Pinene” in the Taubertal variety is 2.5 times higher. This finding emphasizes the importance of genetic factors and the necessity for meticulous planning in crop cultivation and product application.

The data presented in Table 4 clearly illustrate that, aside from the genetic material (plants) utilized, varying cultivation practices can lead to differences in the composition of the essential oil produced, even when the same genotype (variety) is involved. In the context of biostimulant application, it is evident that B2 (comprising seaweeds, auxin at 1.0 mg/lt, and cytokinin at 0.031 mg/lt sourced from Ecklonia maxima) significantly increases the concentrations of “(E)-β-Farnesene”, “Benzyl Benzoate”, “2-β-Funebrene”, “Oxide-Caryophyllene”, and “Spathulenol”; in some cases, almost doubling the concentration, while concurrently reducing the levels of “(E)-β-Ocimene” with a reduction of about 20% compared to the control.

Finally, it is evident from Table 4 that enhanced irrigation leads to a statistically significant increase of approximately 18% in the percentage of “β pinene” when compared to the rainfed treatment. In contrast, the compounds γ-Muurolene, Caryophyllene oxide, (E)-Caryophyllene, δ-Cadinene, and Spathulenol show a statistically significant decrease in their percentages with increased irrigation, exhibiting reductions ranging from 15% to 50% relative to the rainfed treatment.

## 4. Discussion

*Hypericum perforatum* originates from Europe and Asia, where it thrives under specific ecological conditions [35,36]. The rising industrial and medicinal demand for this plant necessitates substantial production volumes [16,37]. Consequently, its cultivation is essential to satisfy both industrial and biotechnological needs. This research provides initial insights into the adaptation of cultivated *Hypericum perforatum* to the climatic and soil conditions found in Greece, a representative Mediterranean nation.

The existing global literature regarding the impacts of cultivating medicinal plants remains insufficient [38,39,40], with particularly limited data available for the Mediterranean region, especially concerning St. John’s Wort. The aim of this study is to initiate a series of experiments.

Most prior research has concentrated on *Hypericum perforatum* development under regulated environmental conditions. For instance, ref. [41] examined the impact of light quality and intensity on the growth of *Hypericum perforatum* L. In a similar vein, ref. [42] investigated how nickel pollution affected *Hypericum perforatum* L. growth and its secondary metabolite profile. Additionally, ref. [43] studied how varying light quality affects growth and secondary metabolite production in the adventitious root cultivation of *Hypericum perforatum*.

Studies contacted in Turkey and in Jordan [44,45,46] recorded the flowering of *Hypericum perforatum* between April and September, which agrees with the results of the current study. One other growth characteristic is plant height, which varies according to different studies. Ref. [47] reported a height range of 45 to 99 cm, whereas [48] indicated that the height can vary between 25 and 44 cm and 55 and 80 cm, depending on the conditions of establishment. The abovementioned range also agrees with the plant height that was found for the different examined factors of this study.

While there is a lack of field studies specifically examining the cultivation of St. John’s Wort and the impact of irrigation and biostimulant use on its yield, the existing literature [49,50,51,52,53,54] indicates a beneficial effect of irrigation on the yields of various aromatic and medicinal plants, similar to the findings of the current research. In the case of biostimulant effects, numerous studies have indicated that the application of biostimulants leads to a notable enhancement in the growth characteristics of various aromatic and medicinal plants, including increases in plant height, leaf count, and both fresh and dry weight such as in the results of the current study [55,56,57,58,59]. Several researchers [60,61,62,63] have suggested that these beneficial effects may stem from enhanced cell elongation, improved cellular membrane permeability, or increased nitrogen absorption, all of which contribute to vigorous root and foliage development. However, it is also noted in the literature that, despite the advantages, the application of biostimulants can lead to a significant decrease in the fresh and dry weights of plants, roots, and chlorophyll b content in certain cases [64].

*H. perforatum* L. essential oil is primarily derived from the aerial parts of the plant [65]. Numerous studies have documented the chemical variability of *H. perforatum*, indicating the presence of chemotypes that lack certain compounds [66,67], which aligns with the findings of the current research. Additionally, this study highlights the significance of plant variety and cultivation methods, which can result in statistically significant variations in both the yield of essential oil and the composition of its various components. The literature [68,69,70] indicates that *H. perforatum* essential oil contains a variety of compounds in significant amounts, including β-pinene (18.32%), α-pinene (5.56–30.92%), δ-Cadinene (0.0–22.58%), and caryophyllene (15.26%), while several compounds are present in lesser quantities, such as (E)-β-caryophyllene, δ-Cadinene, (E)-β-Ocimene, 2-methyldecane, germacrene-δ, and 2-methyl octane, which align with the focus of this research.

The differences in the chemical composition of the essential oils derived from medicinal plants, influenced by several factors as highlighted in this research, represent a major challenge for the modern herbal industry. These compositional differences may restrict the use of specific oils in industrial applications or necessitate their application in alternative products [71,72,73,74,75].

Reference is specifically made to certain substances, including α- and β-pinene, which are significant constituents of the monoterpene family and are frequently present in the essential oils of numerous plants. They are associated with a broad spectrum of pharmacological effects, including modulation of antibiotic resistance, antitumor activity, antimicrobial effects, antimalarial properties, antioxidant capabilities, anti-inflammatory actions, anti-Leishmania effects, and analgesic properties [76]. The diverse biological activities of these phytochemicals facilitate their application in numerous fields, such as fungicides, flavoring agents, fragrances, and both antiviral and antimicrobial products [77]. Additionally, α- and β-pinene are integral components in certain renal and hepatic medications [78]. Their antibacterial properties are attributed to their toxic effects on cellular membranes. Furthermore, studies have indicated that these compounds exhibit inhibitory effects against breast cancer and leukemia [79]. Consequently, higher concentrations of α- and β-pinene in essential oils correlate with enhanced efficacy in these properties. Specifically, treatments that increase α-pinene concentrations include the V2×I2×B1 treatment, while the V1×I2×B3 treatment is effective for β-pinene.

Caryophyllene oxide is an oxygenated terpenoid that typically arises as a metabolic byproduct of Caryophyllene. This compound is recognized for being non-toxic and non-sensitizing, making it a popular choice as a preservative in food, pharmaceuticals, and cosmetics, as well as serving as an insecticide. Its potential application as a cancer treatment is bolstered by findings that indicate a lack of genotoxicity and its effective absorption through cellular membranes. As a result, higher concentrations found in essential oils are associated with improved efficacy in these roles. Notably, treatments that elevate its concentration include V1×I1×B2 and V1×I2×B2.

Finally, a review of the existing literature reveals that Spathulenol exhibits significant bioactive properties, including anticholinesterase activity, antinociceptive and anti-hyperalgesic effects, and anti-mycobacterial properties [80,81,82]. Recent research has also identified Spathulenol as an eco-friendly insecticide effective against aphids [83]. Notably, higher concentrations of Spathulenol found in essential oils correlate with enhanced effectiveness in these applications, with specific treatments such as V1×I1×B2 and V1×I2×B2 leading to increased concentrations.

Therefore, by analyzing the most effective combination of treatments with components at their optimal percentages, it is determined that the treatments V1×I1×B2, V1×I1×B3, and V1×I2×B3 are the most suitable options.

## 5. Conclusions

This study marks the first investigation into the cultivation of the medicinal plant *Hypericum perforatum* under the climatic conditions of a Mediterranean country, specifically Greece. The research focused on the impact of irrigation and biostimulants on two distinct genotypes (varieties) of the plant, assessing both quantitative factors (such as height, drug yield, and essential oil yield) and qualitative characteristics (including essential oil content and composition). The findings highlight the importance of customized irrigation and biostimulant approaches in enhancing both the yield and quality of balsam cultivars during their second growing season, setting a foundation for future advancements in sustainable agricultural methods.

The findings indicate that the genetic material used plays a significant role in the performance of the plant, influencing percentage characteristics such as production potential, height, and flowering, as well as the composition of the essential oil. Notably, one of the two varieties blooms biennially, resulting in either a complete absence of certain compounds in the essential oil or their presence in very limited quantities.

Furthermore, the research revealed that irrigation positively affects the overall yield of the balsam plant, enhancing production while simultaneously reducing the levels of several essential oil compounds. However, a notable increase in β-pinene content was observed during this study.

Regarding biostimulants, their effects appear to vary based on their specific composition, with some contributing to an increase in essential oil compound content while others may lead to a decrease.

These findings represent some of the first data from a field experiment on the cultivation of H. perforatum in a Mediterranean climate. Continued research is essential and could focus on native populations from the surrounding area (to explore different genetic materials), varying irrigation levels, additional biostimulants, fertilization, and other factors that may influence crop yield and alter the quality characteristics, such as the essential oil composition of H. perforatum.

## Figures and Tables

**Figure 1 plants-13-03573-f001:**
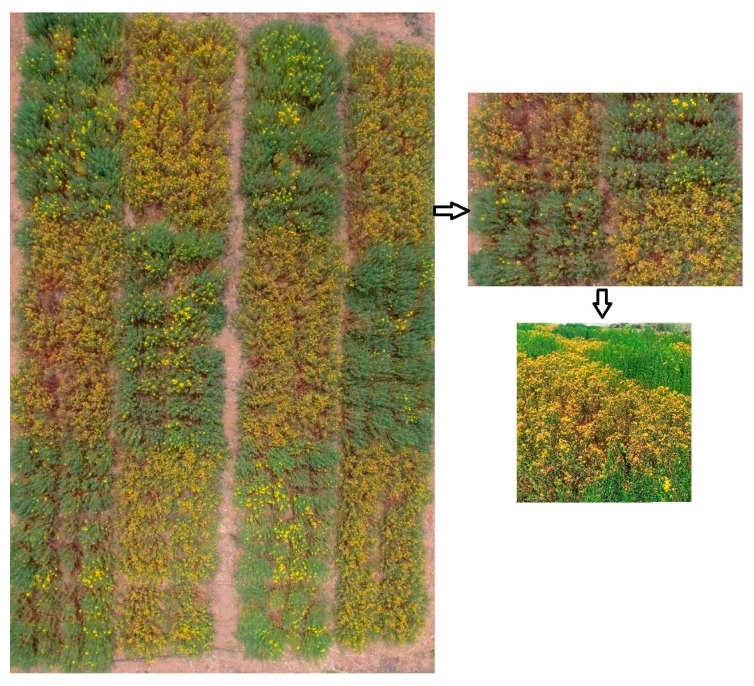
St. John’s Wort treatments.

**Figure 2 plants-13-03573-f002:**
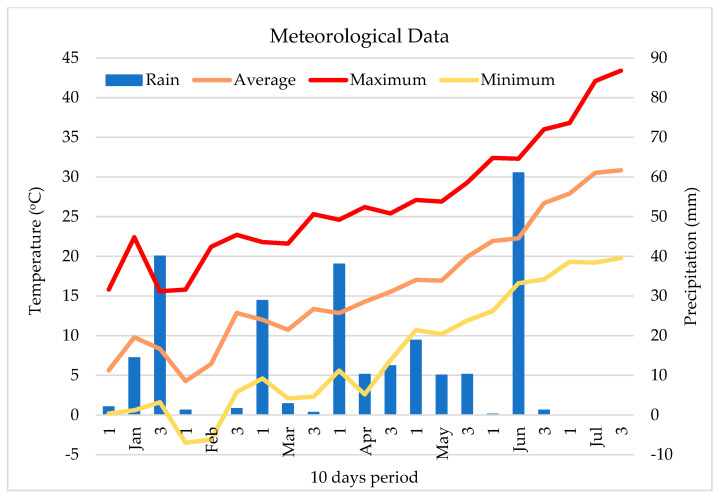
Temperature (Max, Avg, Min) and precipitation from January to July 2023 (10-day intervals).

**Table 1 plants-13-03573-t001:** Plant characteristics (diameter, florescence height, plant height in cm) as affected by different varieties (V1, V2), different irrigation levels (I1, I2), different biostimulants (B1, B2, B3), and their interactions at final harvest.

	Plant Diameter (cm)	Florescence Height	Plant Height (cm)		Plant Diameter (cm)	Florescence Height	Plant Height (cm)
Varieties				Irrigation×Biostimulants			
V1	52.7	55.8	80.7	I1×B1	56.7	44.8	64.2
V2	57.9	47.4	65.5	I1×B2	56.1	45.5	66.4
LSD_0.05_	3.81	5.91	8.15	I1×B3	57.7	45.2	67.0
Irrigation				I2×B1	56.1	58.6	80.9
I1	56.7	45.2	65.9	I2×B2	53.6	57.9	79.6
I2	53.8	58.1	80.3	I2×B3	51.7	57.7	80.4
LSD_0.05_	ns	4.613	4.83	LSD_0.05_	ns	ns	ns
Biostimulants				Varities×Irrigation×Biostimulants			
B1	56.4	51.7	72.5	V1×I1×B1	54.2	45.6	66.9
B2	54.9	51.7	73.0	V1×I1×B2	54.1	48.3	73.2
B3	54.6	51.4	73.7	V1×I1×B3	54.8	47.9	72.1
LSD_0.05_	ns	ns	ns	V1×I2×B1	53.8	66.6	93.6
Varieties×Irrigation				V1×I2×B2	49,8	63,8	87,5
V1×I1	54.3	47.3	70.8	V1×I2×B3	49.6	62.8	90.8
V1×I2	51.0	64.4	90.6	V2×I1×B1	59.2	44.1	61.4
V2×I1	59.1	43.0	61.0	V2×I1×B2	58.1	42.6	59.6
V2×I2	56.6	51.7	70.0	V2×I1×B3	60.2	42.4	61.9
LSD_0.05_	ns	ns	8.59	V2×I2×B1	58.5	50.5	68.2
Varieties×Biostimulants				V2×I2×B2	57.5	52.1	71.7
V1×B1	54.0	56.1	80.2	V2×I2×B3	53.8	52.6	70.1
V1×B2	51.9	56.0	80.4	LSD_0.05_	ns	ns	ns
V1×B3	52.2	55.3	81.4	CV (%)	9.3	11.6	11.9
V2×B1	58.8	47.3	64.8				
V2×B2	57.8	47.4	65.6				
V2×B3	57.0	47.9	66.0				
LSD_0.05_	ns	ns	ns				

LSD: least significant difference; ns: non-significant; V1: Topaz; V2: Taubertal; I1: rainfed; I2: 100% Eto; B1: control; B2: seaweeds, auxin 1.0 mg/lt and cytokinin 0.031 mg/lt from Ecklonia maxima; B3: specific formulation including 10% *w*/*v* of amino acids, 11.3% *w*/*v* of pure protein, 22% *w*/*v* of sucrose, and 10% *w*/*v* of UV filter.

**Table 2 plants-13-03573-t002:** Plant fresh and dry weight as affected by different varieties (V1, V2), different irrigation levels (I1, I2), different biostimulants (B1, B2, B3), and their interactions at final harvest.

	Fr.W (kg ha^−1^)	D.W (kg ha^−1^)		Fr.W (kg ha^−1^)	D.W (kg ha^−1^)
Varieties			Irrigation×Biostimulants		
V1	8371	3137	I1×B1	6285	2497
V2	4522	1775	I1×B2	6286	2407
LSD_0.05_	1598	512.6	I1×B3	5731	2215
Irrigation			I2×B1	7412	2748
I1	6101	2373	I2×B2	6454	2410
I2	6792	2539	I2×B3	6509	2458
LSD_0.05_	ns	ns	LSD_0.05_	ns	ns
Biostimulants			Varities×Irrigation×Biostimulants		
B1	6849	2622	V1×I1×B1	8742	3382
B2	6370	2409	V1×I1×B2	8750	3222
B3	6120	2337	V1×I1×B3	7662	2856
LSD_0.05_	ns	ns	V1×I2×B1	9403	3500
Varieties×Irrigation			V1×I2×B2	7329	2759
V1×I1	8385	3154	V1×I2×B3	8338	3102
V1×I2	8356	3120	V2×I1×B1	3829	1611
V2×I1	3818	1593	V2×I1×B2	3823	1593
V2×I2	5227	1957	V2×I1×B3	3801	1574
LSD_0.05_	ns	ns	V2×I2×B1	5421	1995
Varieties×Biostimulants			V2×I2×B2	5579	2060
V1×B1	9072	3441	V2×I2×B3	4681	1815
V1×B2	8039	2991	LSD_0.05_	ns	ns
V1×B3	8000	2979	CV (%)	35.1	32.6
V2×B1	4625	1803			
V2×B2	4701	1826			
V2×B3	4241	1694			
LSD_0.05_	ns	ns			

LSD: least significant difference; ns: non-significant; Fr.W: Fresh Weight; D.W: dry weight; V1: Topaz; V2: Taubertal; I1: rainfed; I2: 100% Eto; B1: control; B2: seaweeds, auxin 1.0 mg/lt and cytokinin 0.031 mg/lt from Ecklonia maxima; B3: a specific formulation including 10% *w*/*v* of amino acids, 11.3% *w*/*v* of pure protein, 22% *w*/*v* of sucrose, and 10% *w*/*v* of UV filter.

**Table 3 plants-13-03573-t003:** The essential oil content (%) and essential oil yield (kg ha^−1^) as affected by the different varieties (V1, V2), the different irrigation levels (I1, I2), the different biostimulants (B1, B2, B3), and their interactions at the final harvest.

	Essential Oil Content (%)	Essential Oil Yield (kg ha^−1^)		Essential Oil Content (%)	Essential Oil Yield (kg ha^−1^)
Varieties			Irrigation×Biostimulants		
V1	0.128	397	I1×B1	0.133	327
V2	0.124	229	I1×B2	0.087	214
LSD_0.05_	ns	59.2	I1×B3	0.048	115
Irrigation			I2×B1	0.124	345
I1	0.089	219	I2×B2	0.200	468
I2	0.163	407	I2×B3	0.163	408
LSD_0.05_	0.0126	84.8	LSD_0.05_	0.0203	107.7
Biostimulants			Varities×Irrigation×Biostimulants		
B1	0.128	336	V1×I1×B1	0.173	431
B2	0.144	341	V1×I1×B2	0.095	304
B3	0.105	362	V1×I1×B3	0.067	185
LSD_0.05_	0.0145	ns	V1×I2×B1	0.127	443
Varieties×Irrigation			V1×I2×B2	0.165	450
V1×I1	0.096	307	V1×I2×B3	0.185	566
V1×I2	0.159	487	V2×I1×B1	0.138	223
V2×I1	0.082	131	V2×I1×B2	0.078	124
V2×I2	0.166	327	V2×I1×B3	0.028	46
LSD_0.05_	ns	ns	V2×I2×B1	0.122	246
Varieties×Biostimulants			V2×I2×B2	0.235	485
V1×B1	0.127	437	V2×I2×B3	0.142	250
V1×B2	0.130	377	LSD_0.05_	0.0274	137.2
V1×B3	0.126	376	CV (%)	35.1	32.6
V2×B1	0.130	235			
V2×B2	0.157	304			
V2×B3	0.085	148			
LSD_0.05_	0.0187	87.8			

LSD: least significant difference; ns: non-significant; V1: Topaz; V2: Taubertal; I1: rainfed; I2: 100% Eto; B1: control; B2: seaweeds, auxin 1.0 mg/lt and cytokinin 0.031 mg/lt from Ecklonia maxima; B3: a specific formulation including 10% *w*/*v* of amino acids, 11.3% *w*/*v* of pure protein, 22% *w*/*v* of sucrose, and 10% *w*/*v* of UV filter.

**Table 4 plants-13-03573-t004:** Essential oil characteristics (% components found using GC–MS; 2-Methyl-Decane, 2-Methyl-Octane, (E)-β-Farnesene, α-Pinene, Benzyl Benzoate, 2-β-Funebrene, γ-Muurolene, (E)-β-Ocimene, Oxide-Caryophyllene, β-Pinene, (E)-Caryophyllene, γ-Himachelene, δ-Cadinene, Spathulenol) as affected by different varieties (V1, V2), different irrigation levels (I1, I2), different biostimulants (B1, B2, B3), and their interactions at final harvest.

	2-Methyl-Decane	2-Methyl-Octane	(E)-β-Farnesene	α-Pinene	Benzyl Benzoate	2-β-Funebrene	γ-Muurolene	(E)-β-Ocimene	Oxide-Caryophyllene	Β-Pinene	(E)-Caryophyllene	γ-Himachelene	δ-Cadinene	Spathulenol
Varieties														
V1	2.89	22.57	1.43	15.38	1.36	1.13	1.34	1.55	12.90	12.01	8.69	0.98	0.84	5.95
V2	2.46	20.40	0.00	38.26	0.41	0.00	1.38	1.39	3.42	5.34	2.34	0.98	0.82	4.16
LSD_0.05_	ns	ns	0.502	6.393	0.780	0.63	ns	ns	3.147	2.099	1.262	ns	ns	0.341
Irrigation														
I1	2.78	21.92	0.74	25.19	0.94	0.63	1.52	1.87	9.38	7.93	6.29	1.08	1.01	6.83
I2	2.57	21.05	0.69	28.45	0.83	0.50	1.20	1.55	6.94	9.42	4.74	0.88	0.64	3.28
LSD_0.05_	ns	ns	ns	ns	ns	ns	0.224	ns	1.306	1.067	1.113	ns	0.302	0.816
Biostimulants														
B1	2.63	22.73	0.66	28.11	0.73	0.46	1.13	1.60	6.25	8.83	5.69	0.95	0.71	4.76
B2	2.45	19.76	0.85	25.90	1.13	0.74	1.46	1.29	10.91	7.78	6.12	1.03	0.92	5.96
B3	2.94	21.97	0.64	26.45	0.80	0.49	1.50	1.51	7.32	9.41	4.74	0.97	0.84	4.45
LSD_0.05_	0.353	ns	0.094	ns	0.264	0.129	0.244	0.173	1.288	ns	1.105	ns	ns	0.795
Varieties×Irrigation														
V1×I1	2.95	23.99	1.48	15.11	1.48	1.26	1.47	1.64	14.14	10.73	10.01	1.14	0.94	7.34
V1×I2	2.83	21.14	1.37	15.65	1.25	1.00	1.21	1.45	11.67	13.28	7.38	0.83	0.73	4.57
V2×I1	2.61	19.84	0.00	35.27	0.41	0.00	1.56	1.13	4.63	5.14	2.57	1.02	1.08	6.33
V2×I2	2.31	20.96	0.00	41.26	0.42	0.00	1.20	1.64	2.21	5.55	2.1	0.94	0.555	2.00
LSD_0.05_	ns	ns	ns	ns	ns	ns	ns	0.205	ns	1.664	ns	ns	ns	ns
Varieties×Biostimulants														
V1×B1	2.82	22.43	1.313	15.14	1.01	0.92	1.11	1.52	9.51	12.00	8.99	0.93	0.64	6.42
V1×B2	2.37	19.14	1.693	13.75	2.01	1.47	1.78	1.32	18.69	10.30	10.04	1.12	1.19	7.86
V1×B3	3.48	26.14	1.274	17.24	1.08	0.99	1.14	1.80	10.52	13.72	7.06	0.91	0.68	3.58
V2×B1	2.44	23.04	0.00	41.08	0.46	0.00	1.14	1.68	2.99	5.67	2.39	0.968	0.79	3.10
V2×B2	2.53	20.38	0.00	38.05	0.25	0.00	1.14	1.26	3.14	5.25	2.20	0.940	0.65	4.06
V2×B3	2.40	17.79	0.00	35.65	0.53	0.00	1.85	1.22	4.13	5.11	2.42	1.02	1.00	5.33
LSD_0.05_	0.549	ns	0.420	ns	0.592	0.516	ns	0.2146	2.421	1.927	1.414	ns	0.248	0.929
Irrigation×Biostimulants														
I1×B1	2.72	25.4	0.50	26.89	0.73	0.50	1.12	1.52	6.30	9.19	6.68	1.01	0.73	6.64
I1×B2	2.49	19.07	0.89	24.79	1.14	0.83	1.67	1.13	11.82	7.81	6.09	1.08	1.17	7.28
I1×B3	3.13	21.28	0.83	23.88	0.96	0.56	1.77	1.52	10.04	6.8	6.10	1.15	1.13	6.58
I2×B1	2.55	20.07	0.82	29.33	0.74	0.43	1.13	1.68	6.21	8.48	4.70	0.88	0.70	2.88
I2×B2	2.41	20.45	0.80	27.02	1.12	0.64	1.25	1.46	10.01	7.74	6.14	0.98	0.68	4.64
I2×B3	2.75	22.65	0.44	29.01	0.65	0.43	1.23	1.50	4.61	12.03	3.38	0.78	0.56	2.33
LSD_0.05_	ns	ns	0.133	ns	ns	ns	ns	ns	1.771	1.757	1.516	ns	0.321	ns
Varities×Irrigation×Biostimulants														
V1×I1×B1	2.76	27.91	1.00	16.24	0.86	1.00	1.03	1.41	8.30	12.90	10.37	0.84	0.42	8.64
V1×I1×B2	2.35	19.46	1.78	14.36	2.01	1.66	1.91	1.41	18.79	10.71	10.02	1.34	1.37	8.02
V1×I1×B3	3.76	24.62	1.67	14.71	1.56	1.12	1.48	2.10	15.34	8.57	9.63	1.24	1.03	5.34
V1×I2×B1	2.89	16.95	1.63	14.03	1.16	0.85	1.19	1.63	10.72	11.09	7.60	1.01	0.85	4.20
V1×I2×B2	2.39	18.53	1.60	13.14	2.01	1.28	1.64	1.24	18.59	9.88	10.05	0.91	1.01	7.69
V1×I2×B3	3.20	27.65	0.88	19.77	0.59	0.85	0.81	1.49	5.69	18.87	4.49	0.57	0.33	1.82
V2×I1×B1	2.68	22.90	0.00	37.53	0.60	0.00	1.21	1.63	4.29	5.47	2.98	1.18	1.03	4.63
V2×I1×B2	2.62	18.69	0.00	35.21	0.26	0.00	1.42	0.84	4.85	4.91	2.16	0.82	0.97	6.54
V2×I1×B3	2.51	17.95	0.00	33.06	0.36	0.00	2.05	0.93	4.73	5.03	2.57	1.05	1.23	7.81
V2×I2×B1	2.21	23.19	0.00	44.63	0.32	0.00	1.08	1.74	1.69	5.87	1.81	0.75	0.54	1.56
V2×I2×B2	2.43	22.07	0.00	40.89	0.23	0.00	0.86	1.68	1.43	5.60	2.24	1.06	0.34	1.59
V2×I2×B3	2.29	17.64	0.00	38.25	0.70	0.00	1.65	1.51	3.52	5.19	2.27	1.00	0.78	2.84
LSD_0.05_	ns	ns	0.384	ns	0.628	ns	ns	0.326	2.829	2.561	ns	0.686	0.390	1.424
CV (%)	15.3	17.2	15.3	15.8	34.4	26.5	20.7	13.6	18.2	18.3	23.2	24.8	25.1	18.2

LSD: least significant difference; ns: non-significant; V1: Topaz; V2: Taubertal; I1: rainfed; I2: 100% Eto; B1: control; B2: seaweeds, auxin 1.0 mg/lt and cytokinin 0.031 mg/lt from Ecklonia maxima; B3: a specific formulation including 10% *w*/*v* of amino acids, 11.3% *w*/*v* of pure protein, 22% *w*/*v* of sucrose, and 10% *w*/*v* of UV filter.

## Data Availability

The original contributions presented in this study are included in the article. Further inquiries can be directed to the corresponding author.

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
