# Peer review of "Assessing the Impact of Irrigation and Biostimulants on the Yield and Quality Characteristics of Two Different St. John’s Wort Cultivars in Their Second Growing Season"

_plants, 2024, doi:10.3390/plants13243573_

Round 1
Reviewer 1 Report
Comments and Suggestions for Authors
The article entitled “Assessing the impact of irrigation and biostimulants on the yield and quality characteristics of two different St. John's Wort cultivars in their second growing season” describes the value of customized irrigation and biostimulant techniques in raising yield and quality in the second growing season and establishing the foundation for long-term agricultural improvements. Production potential, height, flowering, and essential oil composition were all strongly impacted by genotype one variety showed changed essential oil components and biennial blooming.
I recommend it is highly suitable for publication in the Plants after minor revision. The revisions are,
1) In the introduction part authors should reduce repetition of multiple mentions of the medicinal benefits of St. John’s Wort.
2) For clarity, define all abbreviations upon first use in the text (e.g., "ETo")
3) The authors should clearly mention the irrigation frequency of application.
Author Response
Title: "Assessing the impact of irrigation and biostimulants on the yield and quality characteristics of two different St. John's Wort cultivars in their second growing season"
Dear Editor,
We are thankful for giving us the opportunity to support our work. Following reviewers’ comments/suggestions, we are sending the revised manuscript, based on our feeling that the work has been modified taking into account the comments/suggestions made by the reviewers. In the revised manuscript, we have carried out important changes.
All the changes made in the revised manuscript are in colored yellow. For your decision and comparison with the first version, we present the changes in juxtaposition with reviewer’s suggestions in the next pages.
Sincerely yours,
Kyriakos Giannoulis
Reviewer 1
Comments and Suggestions for Authors
The article entitled “Assessing the impact of irrigation and biostimulants on the yield and quality characteristics of two different St. John's Wort cultivars in their second growing season” describes the value of customized irrigation and biostimulant techniques in raising yield and quality in the second growing season and establishing the foundation for long-term agricultural improvements. Production potential, height, flowering, and essential oil composition were all strongly impacted by genotype one variety showed changed essential oil components and biennial blooming.
I recommend it is highly suitable for publication in the Plants after minor revision. The revisions are,
Comments/Suggestions
In the introduction part authors should reduce repetition of multiple mentions of the medicinal benefits of St. John’s Wort.
Answer
The introduction was modified to minimize the redundancy of several mentions regarding the therapeutic advantages of St. John’s Wort according to reviewers’ comments.
Comments/Suggestions
For clarity, define all abbreviations upon first use in the text (e.g., "ETo").
Answer
All abbreviations defined upon first use in the text according to reviewers’ comments.
Comments/Suggestions
The authors should clearly mention the irrigation frequency of application.
Answer
Irrigation was conducted on a weekly basis, information that added to manuscript according to reviewers’ comments.
Reviewer 2 Report
Comments and Suggestions for Authors
General Observations:
The research addresses a relevant topic regarding the production methodology of medicinal plants. However, the presentation of the results lacks sufficient detail to highlight the most suitable method for obtaining plants with higher therapeutic quality. The discussion is limited and does not thoroughly explore the findings or their practical implications.
Abstract:
Although the importance of adapted irrigation and biostimulants is emphasized, the claims are not supported by the presented results.
The impact of biostimulation on the quality of essential oils remains unclear, and no indication is provided regarding which treatment would be most effective for improving therapeutic quality.
Materials and Methods:
Information about the duration and frequency of treatments with biostimulants is missing.
Soil conditions and moisture levels are not described, making it difficult to assess the impact of fertigation.
Days of treatment before harvest and differences in flowering time between varieties are not provided.
Results:
The description is unclear and does not highlight significant differences between treatments.
It is recommended to simplify and clarify the statistical results, relegating the statistical tables to supplementary material.
The description of the compounds measured in the essential oils is exhaustive, but the relevance of the treatments to the composition of the essential oils is not indicated.
Discussion:
There is no clear analysis of which treatments are most advantageous for improving cultivation methods and their potential application in commercial production.
General Conclusion:
The article requires revision to:
- Clarify the methodology, ensuring that experimental conditions and key variables are fully described.
- Present the results more concisely, highlighting statistically significant findings.
- Expand the discussion to emphasize the practical and therapeutic implications of the evaluated treatments.
- Identify the most beneficial treatment and justify its relevance in terms of therapeutic quality.
Author Response

(The authors gave the same response as above.)

Reviewer 3 Report
Comments and Suggestions for Authors
Authors aims to cultivate Hypericum perforatum in Greece under different cultivation practices and to examine how irrigation and the use of biostimulants can affect their overall performance during the second growing season of two St. John's Wort varieties. The findings highlighted the significance of tailored irrigation and biostimulant approaches in enhancing yield and quality during the second growing season. Genotypes significantly influenced production potential, height, flowering, and essential oil composition. Irrigation positively affected yield, but reduced certain essential oil compounds, while increasing β-pinene content. Biostimulants' effects varied depending on composition, with some increasing and others decreasing essential oil compound content. The results are meaningful, but it cannot be accepted in the status.
1. The experiment design is simple and deeper mechanism work is suggested.
2. Why choose irrigation and biostimulants to improve yield and quality of St. John's Wort? What’s the main agricultural practice in improving the yield and quality? Are there any problems or disadvantages to the practice existing? The reason described in the section Introduction now is not enough.
3. The plant figures of St. John's Wort treatments and the control are needed.
Author Response

(The authors gave the same response as above.)
